# Signalling, Information and Consumer Fraud

## Silvia Martinez-Gorricho 

Department of Economics, Universidad Católica de la Santísima Concepción, Alonso de Ribera,
Concepción 2850, Chile; smartinez@ucsc.cl

**Abstract:** In a two-sided asymmetric information market, the role of the accuracy of consumers' imperfect and private information on the level of fraud, incidence of fraud and trade under price rigidity is examined. Consumers receive a costless but noisy private signal of quality. The product offered in the market can be of two exogenously given qualities and it is common knowledge that the consumer is not willing to pay a high price for a low quality product. A low quality seller chooses to be either honest (by charging the lower market price) or dishonest (by charging the higher price). We show that equilibria involving fraud exist for all parameter values. Furthermore, for some parameter values, we find that -in equilibrium- a higher precision of consumers' private information leads to higher levels of fraud and incidence of fraud, reducing consumers' welfare. We provide conditions for the public revelation of consumers' private information to be a Pareto improvement.

**Keywords:** consumer fraud; incomplete information; information structure; price signalling

## 1. Introduction

> *The market is the place set apart where men may deceive one another.*
>
> *Anacharsis, 600 B.C.*

Consumer fraud, a consequence of consumers' incomplete information, is ubiquitous. As an illustration, "seafood fraud" is frequent in the international seafood industry: less desirable, cheaper or more readily available fish are often mislabelled as more desirable species for financial gain. For instance, Oceana's 2018 report entitled "Seafood Fraud and Mislabelling Across Canada" revealed over 44 per cent of mislabelled samples. Oceana's investigations in Halifax, Ottawa, Toronto, Vancouver and Victoria found cheaper haddock and pollock substituted for cod, farmed salmon served up as wild salmon, and escolar (a fish banned in many countries because of its health risks) masqueraded as butterfish or white tuna.[1]

Information provision is considered a market response to the consumers' incomplete information. Some examples are information provided by independent third parties such as consumer report magazines, consumer opinion websites and institutional warnings. A vast number of Consumer Protection Agencies, listed by the U.S. Federal Trade, have been created worldwide as a consequence. These agencies periodically issue consumer warnings alerting the public to potentially misleading activities in the marketplace. For instance, the Australian Competition and Consumer Commission is currently warning Australians about COVID-19-related scams, including the selling of products that claim to cure it or prevent its contagion. The purpose of all these warnings is to help consumers detect such potentially misleading activities or devices in order to avoid becoming a victim of fraud.

---

[1] Refer also to 2019 Oceana's report "Casting a Wider Net: More Action Needed to Stop Seafood Fraud in the United States" and 2016 Oceana's report "Deceptive dishes: Seafood swaps found worldwide". All Oceana's reports are available at: http://usa.oceana.org/publications/reports.

Consumers rationally take into account imperfect signals of quality when taking decisions. If the information privately collected by consumers is informative but noisy, then two-sided information asymmetries exist in the market since both suppliers and consumers are unable to identify the type of the market participant they face. A priori, the more descriptive or detailed the agencies' warnings are, the higher the consumers' ability to identify potential frauds is. This ability can be interpreted as the precision of a private noisy signal observed by consumers about the quality of the offered product. Since warnings are public information, sellers can use them to deduce consumers' said ability.

Our theoretical development is motivated by this evidence. Our major objective is to determine the role played by the precision of consumers' private information on the equilibrium level of fraud and incidence of fraud under price rigidities. Our results suggest that the relationship between these variables is non-monotonic.

We introduce a simple price-quality signalling model of an experience good with private information on both sides of the market. The -exogeneously given- quality of the good is either high or low. The seller knows the quality of his good and he requests one of the two exogenously given market prices (low or high). Consumers are ex-ante homogeneous. A consumer observes the price demanded by the seller and a private noisy binary signal of quality and chooses whether to purchase the good. The consumers' information structure determines the probability that the signal reveals the state. Quality can also be partially inferred from prices. Price rigidity allows consumers to capture some gains from trade if the fixed market prices are set below the consumers' valuations for the goods.[2] Therefore, there are potential gains from trade to be realized in each state of nature. However, we assume it is common knowledge that no consumer is willing to pay the high price for a low quality product. Henceforth, selling a low quality product at the high price will be referred to as fraud.[3]

The market informational asymmetries create incentives for opportunistic behaviour by a seller with a low quality good could attempt to defraud the buyer by charging the high price. However, his probability of success in deceiving the buyer is dependent on the consumer's ability to detect an attempt of fraud. Therefore, the market level of attempted fraud and the incidence of fraud naturally depend on the information structure of the buyer. Our first finding shows that equilibria involving fraud exist for all possible signal precision levels. We fully characterize the market equilibria in terms of two interaction factors: the precision of the buyer's private information and the prior probability that the seller is of a high type. The uniqueness of fraudulent equilibrium outcomes is guaranteed for generic values of both parameters.

In this context, it seems intuitive to conjecture that an increase in the precision of the buyer's private information reduces the probability the buyer is deceived. After all, if the buyer becomes better informed about the quality of the object for sale, the seller will be less successful in misrepresenting it and he will be able to defraud the buyer less frequently. This lower probability of trade should restrain the seller from overcharging and induce him to be more honest. However, this argument is incomplete. Our second finding shows that a higher precision of consumers' private information leads to higher levels of fraud and incidence of fraud for some parameter values in equilibrium. For sufficiently low priors that the good is of high quality, no buyer is willing to purchase the good at the high price if the signal is sufficiently noisy under outright cheating. In this region, the low-quality seller randomizes targeting only the high-signal buyer whom he confronts relatively often. If prices reveal partially the quality of the good, as the private signal's precision increases, a buyer who observes a high signal and a high price becomes more optimistic about the product's quality while a buyer who receives a low

---

[2]　With flexible prices, better information provision helps consumers make better informed choices but at the same time, it deprives consumers of informational rents so that the benefit to consumers of making a more informed choice may be outweighed by the price distortion (refer to [1]).

[3]　In the English dictionary by Oxford University Press, fraud is defined as the action or an instance of deceiving somebody in order to make money. Alternatively, it is defined as a thing that is not what is claimed to be.

signal and a high price becomes more pessimistic about its quality. Since the high-signal buyer becomes more convinced of the product's high quality as the signal's precision increases, her higher willingness to pay for the product induces the low-quality seller to attempt fraud more often, increasing the incidence of fraud. In that case, a more precise consumer's information benefits the seller but harms the buyer. These results suggest that the empowerment of consumers by public policies favouring accurate information provision may go against the goal of enhancing consumers' welfare.

Comparative statics are also performed on the price caps. A higher low price increases the acceptance rate by the marginal buyer in partial-pooling equilibrium outcomes, increasing the market incidence of fraud and reducing consumers' welfare. A higher high price has two counteracting effects: on one hand, it disciplines low-quality sellers but on the other hand, it aggravates the cost of fraud to the consumers whenever it is committed and reduces the consumer's surplus of trading high-quality items. It turns out that the first (second) effect dominates for sufficiently imprecise (precise) signals, increasing (reducing) consumers' welfare.

Finally, we show that the non-monotonicity of fraud in the precision of the private information: (i) is robust to the public revelation of consumers' private information or the costly acquisition of perfect information; and (ii) crucially depends on the dissemination of information. To prove the latter result, we examine more extreme information structures, often analysed in the literature, in which an exogenous fraction of consumers have access to perfect information about the quality of the good on sale while the rest of consumers remain uninformed. It can be shown that the equilibrium incidence of fraud is a non-increasing function of the fraction of informed consumers in the market. This result suggests that the relationship between information asymmetries and the equilibrium incidence of fraud depends on how information is sorted among consumers.

The paper is organized as follows. Section 2 provides a literature review. The formal model and some preliminaries are introduced in Sections 3 and 4 respectively. Section 5 carries out the equilibrium analysis and presents the central results of the paper. The value of information and regulation policies are calculated in Section 6. Section 7 provides a discussion, extending the results of the model in several directions. Section 8 concludes the paper. Proofs are contained in Appendix A.

## 2. Literature Review

The existing work of the literature on price as a signal of quality can be embodied into two main areas of research: a first area is concerned with the moral hazard aspects of the choice of quality, while the second focuses on the classical adverse selection problem, pioneered by [2]. The latter line of research is the one pursued in this paper. In these models, quality is not treated as a choice variable but, instead, is exogenously given. A great majority of the papers that belong to this body of literature assume the potential customers to be either exogenously perfectly informed or completely uninformed about the quality of the good on sale. Several papers ([3–7]) relax this assumption. For instance, Ref. [3] model endogenous information acquisition in which buyers can choose to collect information at some cost and become either perfectly informed or completely uninformed ex-post.

Our modelling approach is in the spirit of [6] (work): we specify that each consumer observes a private and imperfect binary signal without incurring any cost. As a result, the model is categorized as a two-sided asymmetric information model. The central question addressed by the price-quality signalling models consists on determining whether in such settings, prices alone are capable of conveying information on quality in equilibrium. Therefore, Ref. [6] restrict attention to pure strategy sequential equilibria and investigate the existence of fully separating equilibrium outcomes that survive selection criteria. Thus, in such equilibrium outcomes, honest reporting prevails in the market. In contrast, we focus on pooling and mixed strategies, which may exhibit various degrees of fraud. Furthermore, we perform comparative statics in terms of different parameters of interest in order to determine whether more or less information is revealed in equilibrium, and calculate its impact on the level of fraud in the market.

Ref. [7] consider a two-sided asymmetric information model in which for sufficiently poor quality goods, there are not gains of trade as the monopolist's valuation for these goods is higher than the buyer's valuation. The authors find that in the only equilibrium outcome that survives criterion D1 trade always collapses. The spiralling of prices (and thus, market breakdown) could be avoided if the monopolist were able to fully commit ex-ante (before learning his type) to a price cap. In this setting, price rigidity may promote trade and the informativeness of equilibrium prices.

An alternative strand of the literature closely related to our paper is the literature on the value of information. The closest antecedent to our work in this paper is [8] who use a model of binary states and binary signals in which two sellers, with perfectly negatively correlated qualities, compete on (flexible) prices for an informed buyer. The authors address the issue of the value of private and public information respectively by performing comparative statics with respect to the precision of the private information on the demand side and the prior belief. In their monopoly benchmark, they focus on analysing the monopolist's profit function which is found non-monotonic in the precision of the buyer's signal: for sufficiently low precision values, pooling becomes optimal but the monopolist is forced to reduce the selling price in order to keep capturing the buyer with the unfavourable signal. For sufficiently high precision values, posting a separating price becomes optimal. Instead, we find that if the high price is greater than the ex-ante valuation of the good, then for sufficiently low (high) precision values, the low-quality seller randomizes in a mixed-strategy equilibrium in which the buyer with the favourable (unfavourable) signal becomes the marginal buyer.

Other related articles that belong to the literature on the value of information also belong to the literature on quality uncertainty. In these models, sellers are uncertain about the quality of their products and as a result, there is no scope for signalling ([9]). For instance, Ref. [10] finds that the relationship between information asymmetries and trade (measured by the maximum probability with which a good can be traded) is non-monotonic in a competitive market characterized by one-sided information asymmetries (only the sellers have access to an imperfect private information). This result is consistent with our findings in a completely different setting in which the seller is perfectly informed about the quality of his product and a price setter. Ref. [11] analyses a non-linear pricing model categorized as a two-sided asymmetric information model. The monopolist can choose to commit to publicly reveal his private information. The authors show that the monopolist is always better off by committing to reveal directly the information inferred by the buyer in equilibrium and to destroy the remaining information. Finally, Ref. [12] analyses a model in which quality information is public and hence, there are no information asymmetries. The author identifies two properties of the cost functions that lead to a negative value of information for consumers: increasing returns to scale and "sufficiently" convex marginal costs.[4] We demonstrate that the result that consumers may sometimes prefer less public information about product quality also extends to a signalling model with information asymmetries. Furthermore, the author also provides an example where private information about quality may hurt consumers facing a monopolist price setter. However, his example is constructed under the assumption that trade is not desirable in the low-quality state under full information. We obtain a similar finding in a signalling setting under the assumption that trade is always desirable under full information.

There is some previous work on the issue of fraud in the literature on credence goods ([13,14]). With credence goods, consumers cannot judge actual quality either before or after purchase. Under a liability assumption, consumers are protected from undertreatment and if the cost of treatment is increasing in quality, overcharging is strictly more profitable than overtreatment. Consequently, an expert who faces a consumer with a minor problem explicitly chooses to be honest by reporting the minor problem and charging for the inexpensive treatment or to be dishonest by reporting the minor problem as serious and charging the expensive treatment but providing the inexpensive one.

---

[4]　　It is assumed that the cost of production is independent of quality and it is strictly increasing in quantity.

Ref. [15,16] identify equilibria in which sellers mislead some of their customers in a regulated market with fixed prices. Ref. [17] finds that identifiable customer heterogeneities play a crucial role in explaining expert cheating: experts replace price discrimination by cheating their customers selectively. Ref. [18] introduces a model in which a fraction of consumers receive a public noisy but informative signal about the severity of their problem. Consumers are also heterogeneous in terms of their valuation for getting a serious problem fixed. They find that the expert's ability to set prices serves as a substitute for opportunistic cheating. Finally, Ref. [19] shows under non-identifiable customer heterogeneities that less-than-fully expert consumers can benefit from withholding, rather than disclosing, some of their level of expertise from the seller. We consider an extension in which the consumers' private information is revealed publicly. As in [17,18], sellers tailor their cheating behaviour in this one-sided asymmetric information model according to the type of signal publicly observed by the consumer. We characterize the existing equilibria in this one-sided asymmetric model and compare their outcomes to the equilibrium pooling and mixed strategies of the two-sided asymmetric information model. We identify the parameter configurations for which the public revelation of the consumers' private information is a Pareto improvement.

## 3. The Model

Consider a market for an experience good for which quality is the only characteristic relevant to a buying decision. There is a seller with one good for sale and a potential buyer with a unit demand.[5] Two exogenously given qualities are offered in the market. It is common knowledge that the good is of either high quality ($\theta = H$) with probability $\pi \in (0, 1)$ or low quality ($\theta = L$) with probability $1 - \pi$. Henceforth, we will refer to $\pi$ simply as the prior. The seller and consumer differ from each other in their valuation for different quality items. The consumer values the good of quality $\theta$ at $v_\theta$ with $0 < v_L < v_H$. Let $\bar{v}$ denote the buyer's ex ante expected valuation for a given good: $\bar{v} = \pi v_H + (1 - \pi) v_L$. The seller's reservation value for both types of items is normalized to zero. All agents are risk neutral. The seller maximizes expected revenue and the buyer maximizes the expected valuation net of the price paid. Contrary to [7], there are positive potential gains from trade for all quality types.

The seller knows his actual, realized quality, but his potential customer does not and there is no credible direct way by which the seller can provide this information before the customer makes her purchasing decision. However, the seller cannot completely disguise the true quality of his product. Assume that prior to purchase, the consumer obtains without cost a private binary signal, $s$, which conveys a certain amount of information about product quality. We denote the two signal realizations by $s = h$ (high) and $s = l$ (low). We consider a symmetric binary signal structure represented by a parameter $\delta \in \left(\frac{1}{2}, 1\right)$. The buyer observes signal $h$ with probability $\delta$ if the true quality is $H$, and with probability $1 - \delta$ if the true quality is $L$. The number $\delta$ is interpreted as the precision of the consumer's signal. In the limit, when $\delta = 1$, quality can be deduced with certainty by pure inspection before consumption. This is the case of symmetric information. In the other extreme case, if $\delta = \frac{1}{2}$, the signal would become totally uninformative corresponding to the case of one-sided asymmetric information. In our model, the signal is imperfectly correlated with the true quality of the product so it is a two-sided asymmetric information model.

We study the case in which price is the only signalling variable available to the seller. The seller sets a take-it-or-leave-it price $p \in \{p_L, p_H\}$ for his unit, knowing its quality. Prices are exogenously fixed and satisfy $0 < p_L \leq v_L < p_H < v_H$.[6] As a result, it is common knowledge that the buyer is

---

5    This is equivalent to assuming that there are multiple buyers who do not interact strategically with each other.
6    If prices are endogenous, then a plethora of equilibria emerge in a one period monopoly setting and among the refinements suggested in the literature, only criteria D1, which is equivalent to Universal Divinity and Never a Weak Best Response in this setting, has any power in pruning the set of outcomes, and its power is very limited.

not willing to purchase a low quality item at the high price under full information. Otherwise, trade is desirable.

**Definition 1.** *"Fraud" is the act of selling a low-quality good at the high price, which exceeds the consumer's willingness to pay for the low-quality good under full information.*

The consumer is Bayesian rational; she has beliefs over the seller's types and she uses all the available information to update her beliefs according to Bayes rule. The buyer's strategy is simply whether to accept or reject the offer proposed by the seller. If the price offer $p$ proposed by the seller is accepted by the buyer, the buyer consumes the good and she enjoys its true valuation. The $\theta$-quality seller's and the buyer's (ex post) payoffs if the offer at price $p$ is accepted are given by $R(\theta, p) = p$ and $u(\theta, p) = v_\theta - p$ respectively, where $v_\theta \in \{v_L, v_H\}$. If the offer is rejected, the buyer does not consume the good and both agents' payoffs are zero.

## 4. Preliminaries

Let $\phi_\theta$ denote the seller's strategy: the probability that a seller with a good of quality $\theta$ charges the high price for all $\theta \in \{L, H\}$. Let $b(p, s)$ denote the buyer's strategy: the probability that a buyer who observes signal realization $s$ and is offered the good at price $p$ accepts the offer proposed by the seller, for all $s \in \{l, h\}$ and $p \in \{p_L, p_H\}$.

We confine attention to (weak) Perfect Bayesian Equilibrium (PBE). A Perfect Bayesian Equilibrium (PBE) consists of beliefs and strategies satisfying the following requirements: (i) given the players' beliefs, their strategies are sequentially rational; (ii) at each information set, the player with the move has a belief about which node in the information set has been reached by the play of the game. At information sets on the equilibrium path, beliefs are determined by Bayes' rule and the players' equilibrium strategies.

In the limiting case of perfect symmetric information ($\delta = 1$), a unique separating equilibrium exists. The buyer's optimal strategy is to accept trade at the high price if and only if she obtains a high signal realization and to accept trade at the low price for every possible signal realization. As a result, the seller's unique best reply to the buyer's strategy is to be honest, by setting the low price if he has a low quality good and vice versa. Trade occurs with certainty in equilibrium so that the expected gains from trade are fully realized.

Once the potential customer observes the price-signal pair $(p, s)$, she updates her beliefs as to which type of seller she faces: $\mu(H|p, s)$ denotes her posterior beliefs that she is confronting a high-type seller. The potential customer's optimal decision is to purchase the good if and only if the price posted by the seller does not exceed the ex-post expected valuation of the good, which is given by $v(p, s) := v_L + \mu(H|p, s)(v_H - v_L)$. As a result, for all $s \in \{l, h\}$, the buyer's optimal trading decision rule takes the form:

$$b^*(p_L, s) = 1$$

$$b^*(p_H, s) = \begin{cases} 1 & \text{if } \mu(H|p_H, s) > \frac{p_H - v_L}{v_H - v_L} \\ [0, 1] & \text{if } \mu(H|p_H, s) = \frac{p_H - v_L}{v_H - v_L} \\ 0 & \text{if } \mu(H|p_H, s) < \frac{p_H - v_L}{v_H - v_L} \end{cases} \tag{1}$$

where $(p_H - v_L)/(v_H - v_L) \in (0, 1)$ could be interpreted as a proxy for the relative (potential) cost of fraud to the buyer.

Upon having received a high price offer, the posterior beliefs of the customer who observes the high signal realization are at least as high as the posterior beliefs of the customer who observes the low signal realization: $\mu(H|p_H, h) \geq \mu(H|p_H, l)$. This implies $b^*(p_H, h) \geq b^*(p_H, l)$ and therefore, $\delta b^*(p_H, h) + (1 - \delta)b^*(p_H, l) \geq (1 - \delta)b^*(p_H, h) + \delta b^*(p_H, l)$. Thus, although in this model, there does not exist an explicit cost of signalling high quality by charging the high price, there exists an implicit opportunity cost in terms of the probability of trade. As in the standard signalling games,

this opportunity cost is (weakly) higher for the low quality seller than for the high quality seller. As a result, it must be the case that the high quality seller charges the high price in equilibrium with a probability at least as high as the probability at which the low quality seller charges this price: $\phi_H^* \geq \phi_L^*$. If no customer type accepts the high price offer, then the sellers' optimal response is to pool on the low price. If all customer types always accept the high price offer, then the sellers' optimal response is to pool on the high price. This simple observation allows us to begin by stating the following lemma:

**Lemma 1.** *Suppose $\frac{1}{2} < \delta < 1$. An equilibrium is of one of the following types:*

1. $\phi_H^* = \phi_L^* = 0$.
2. $\phi_H^* = 1$ and $\phi_L^* \in (0,1]$.

The main focus of this paper is analysing the effect of information on the level of consumer fraud in equilibrium. We consider the candidate pooling equilibria in which both types of sellers charge the low price not interesting since there is no fraud attempted in such potential equilibria. Moreover, these potential equilibria can be shown to fail an extension of criterion D1 to a setting with two types of receivers and the reasonable assumption about out-of-equilibrium beliefs $\mu(H|p_H,h) \geq \mu(H|p_H,l)$. Consequently, we restrict our analysis to study only the remaining candidate equilibria. In those potential equilibria, the high quality seller always offers his item for sale at the high price while the low quality seller may pool on the high price or may randomize between both prices. Completely fraudulent behaviour by the low quality seller ($\phi_L^* = 1$) is encompassed in pooling equilibria whereas partial honest behaviour by the low quality seller ($\phi_L^* \in (0,1)$) is revealed in hybrid equilibria. The low-quality seller faces the trade-off between obtaining a low revenue surely or a high revenue with a probability strictly lower than one.

**Definition 2.** *A Fraudulent Pooling Equilibrium is a PBE in which both sellers pool on the high price ($\phi_H^* = \phi_L^* = 1$). A Fraudulent Hybrid Equilibrium is a PBE in which the high-quality seller charges the high price while the low-quality seller randomizes between charging the high price and the low price ($\phi_H^* = 1$ and $\phi_L^* \in (0,1)$).*

## 5. Equilibrium Analysis

For any given pair of strategies of the form $\phi_H = 1$ and $\phi_L \in (0,1]$ by the seller, the buyer's posterior beliefs after observing $(p_H, s)$ are given as follows by Bayes' rule:

$$\mu(H|p_H, s; \phi_L) = \begin{cases} \left(1 + \left(\frac{1-\pi}{\pi}\right)\left(\frac{1-\delta}{\delta}\right)\phi_L\right)^{-1} & \text{if } s = h \\ \left(1 + \left(\frac{1-\pi}{\pi}\right)\left(\frac{\delta}{1-\delta}\right)\phi_L\right)^{-1} & \text{if } s = l \end{cases} \tag{2}$$

Figure 1 displays the buyer's posterior beliefs after being offered the item at the high price as a function of the precision of her private signal and the strategy of the low-quality seller.

Due to the informativeness of the private signal, the posterior beliefs upon receiving a high (low) signal realization and a high price are increasing (decreasing) in the precision level of the signal. Thus, if the high price is posted, the difference in the expected value of the good between a buyer who observes the high signal realization and a buyer who observes the low signal realization, $v(p_H, h) - v(p_H, l)$, is increasing in the accuracy of the private information $\delta$ (e.g., the posterior beliefs have the monotone likelihood ratio property).

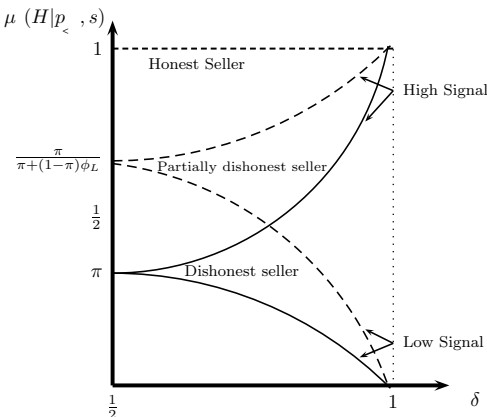

**Figure 1.** The buyer's posterior beliefs after receiving a high price offer as a function of her signal realization $s$, the signal accuracy $\delta$ and the low-quality seller strategy $\phi_L$, if $\pi < \frac{1}{2}$ and $\phi_H = 1$.

Similarly, the low-quality seller's strategy corresponds to the precision of the prices as signals of quality. The lower is the probability of attempting fraud by the low-quality seller, the more informative is the public signal set strategically by the seller, and the more optimistic become both types of customers about the quality of the good on sale after receiving a high price offer. As a result, $v(p_H, h) - v(p_H, l)$ is increasing in $\phi_L$.

We define $\delta_h := (1 + (\pi/(1 - \pi))((v_H - p_H)/(p_H - v_L)))^{-1}$ as the signal precision level at which the buyer who observes the high signal realization is indifferent between accepting or rejecting the high price offer under outright cheating if $p_H > \bar{v}$. Hence, it satisfies $v(p_H, h) = p_H$ for $\phi_L = 1$. In such case, the low-signal customer cannot be made indifferent since then $\mu(H|p_H, l; \phi_L) < \pi < (p_H - v_L)/(v_H - v_L)$ for $\phi_L = 1$. Similarly, $\delta_l := 1 - \delta_h$ denotes the signal precision level at which the buyer who observes the low signal realization is indifferent between accepting or rejecting the high price offer under outright cheating if $p_H < \bar{v}$. Thus, it satisfies $v(p_H, l) = p_H$ for $\phi_L = 1$. If $p_H \leq \bar{v}$, the high-signal customer cannot be made indifferent since then $(p_H - v_L)(v_H - v_L) \leq \pi < \mu(H|p_H, h; \phi_L)$ for all $\phi_L \in (0, 1]$ and $\frac{1}{2} < \delta < 1$.

Next result establishes conditions for the existence of fraudulent pooling and hybrid equilibria. Its proof, located in the Appendix, characterizes the different type of equilibrium outcomes that can be supported for the different parameter configurations.

**Proposition 1.** *Assume* $\frac{1}{2} < \delta < 1$.

   (i) *If* $p_H > \bar{v}$ *and* $\delta_h > 1 - \frac{p_L}{p_H}$, *no fraudulent pooling equilibrium exists while fraudulent hybrid equilibria exist for all* $\delta \in \left(\frac{1}{2}, 1\right)$.

  (ii) *If* $p_H > \bar{v}$ *and* $\delta_h \leq 1 - \frac{p_L}{p_H}$, *then*

      (a) *Fraudulent pooling equilibria exist if and only if* $\delta \in \left[\delta_h, 1 - \frac{p_L}{p_H}\right]$.
      (b) *Fraudulent hybrid equilibria exist if and only if* $\delta \in \left(\frac{1}{2}, \delta_h\right) \cup \left[1 - \frac{p_L}{p_H}, 1\right)$.

  (iii) *If* $p_H \leq \bar{v}$ *then*

      (a) *Fraudulent pooling equilibria exist if and only if* $\delta \leq \max\left\{\delta_l, 1 - \frac{p_L}{p_H}\right\}$.
      (b) *Fraudulent hybrid equilibria exist if and only if* $\delta \geq \max\left\{\delta_l, 1 - \frac{p_L}{p_H}\right\}$.

Proposition 1 implies the following result.

**Corollary 1.** *Equilibria involving fraud exist for all* $\delta \in \left(\frac{1}{2}, 1\right)$.

Proposition 1 has three important implications regarding existence of equilibria: (i) pooling on the high price can be supported in equilibrium only if the prior is sufficiently high; (ii) partial dishonest behaviour by the low quality seller can be supported in equilibrium for all prior levels; (iii) fraudulent equilibria exist for all parameter values (Corollary 1). To see the former result (i), note that the inequalities $p_H > \bar{v}$ and $\delta_h > 1 - (p_L/p_H)$ outlined in Proposition 1 are simultaneously satisfied only for sufficiently low prior values since the higher is the prior, the greater is the ex-ante expected valuation of the good and the lower is the indifference precision threshold for the buyer who observes a high-signal realization ($\delta_h$). Thus, the argument for result (i) is as follows: if the prior is sufficiently low, the buyer who observes the low-signal realization does not accept a high-price offer as both the ex-ante and ex-post valuations of the good fall short of the high-price under outright cheating (refer to Figure 1). The buyer who observes the high-signal realization would be tempted to purchase the good if and only if the signal precision exceeded her indifference threshold $\delta_h$. However, if so, the probability of a match with this buyer would be so low for a low-quality seller that he would find it more profitable to be honest and charge the low price ($p_L > (1 - \delta_h)p_H$). Therefore, if the high price is above the ex-ante expected valuation of the good ($p_H > \bar{v}$), a necessary and sufficient condition for the existence of pooling equilibria is an intermediate signal precision ($\delta \in \left[\delta_h, 1 - \frac{p_L}{p_H}\right]$) as the signal must be sufficiently accurate for the buyer who observes a high-signal realization to be willing to purchase the good under outright cheating ($\delta \geq \delta_h$) but of bounded precision for dishonest behaviour to be profitable ($\delta \leq 1 - (p_L/p_H)$).

Uniqueness of fraudulent equilibrium outcomes is guaranteed for almost all values of $\delta$. For the non-generic cases $\delta = \delta_h$ and $\delta = \delta_l$, there exists a continuum of pooling equilibrium outcomes parametrized by the randomization strategy of the customer who observes the high and low signal realization respectively. For the non-generic case $\delta = 1 - (p_L/p_H)$, there exists a continuum of hybrid equilibrium outcomes in accordance with the seller's randomization strategy, which is not pinned down but bounded from above and below.

Further, we restrict our discussion of the existence conditions outlined in Proposition 1 to the interesting parameter configuration $p_H > \max\{\bar{v}, 2p_L\}$ since it implies the non-monotonicity of fraud in consumers' private information, a result shown in next section (Proposition 2). We can distinguish three different regions according to the accuracy of the private signal that lead to different equilibrium regimes. If the consumer's private information is sufficiently imprecise ($\delta < \min\{\delta_h, 1 - (p_L/p_H)\}$) no pooling equilibrium exists by the argument mentioned above. Instead, a fraudulent hybrid equilibrium outcome in which the low-quality seller randomizes between both prices extracting the high-signal buyer's full surplus can be supported in equilibrium in this region of bounded precision. We call this equilibrium outcome "fraudulent hybrid H" equilibrium outcome since the high-signal buyer is the marginal or "targeted" buyer. If the private information is of intermediate precision ($\delta \in (\delta_h, 1 - (p_L/p_H))$), the high-signal consumer becomes so sufficiently optimistic about the quality of the good that she is willing to purchase the good at the high price even in the most pessimistic scenario of outright fraud, receiving an informational rent. As the probability of a match with a high-signal buyer is sufficiently high for the low-quality seller, it is optimal for him to practice outright fraud. Thus, the "public" signal set strategically by the seller becomes uninformative and the customer relies exclusively on her private information to deduce the quality of the object on sale. We call this equilibrium outcome "fraudulent pooling H". Finally, if the consumer's private information is sufficiently precise ($\delta > 1 - (p_L/p_H)$), the low-quality seller is partially honest in equilibrium: the low-signal buyer becomes the marginal buyer when choosing his pricing randomization strategy since the probability of meeting a high-signal buyer is too low. We call this equilibrium outcome "fraudulent hybrid L" equilibrium outcome.

Remark 1 displays the partial characterizations of the above mentioned fraudulent equilibria which are relevant in the sequel.

**Remark 1.** *Assume $\delta \in \left(\frac{1}{2}, 1\right)$. Two different types of fraudulent hybrid equilibria can be distinguished for generic values of $\delta$:*

(i) *A fraudulent hybrid L equilibrium is characterized by $\phi_H^* = 1$, $\phi_L^* = \left(\frac{\pi}{1-\pi}\right)\left(\frac{1-\delta}{\delta}\right)\left(\frac{v_H - p_H}{p_H - v_L}\right) :=$*
   *$\underline{\phi}(\delta, \pi, p_H), b^*(p_H, h) = 1$ and $b^*(p_H, l) = 1 - \frac{1}{\delta}\left(1 - \frac{p_L}{p_H}\right)$.*
   *The ex-ante equilibrium payoffs of the seller and the buyer are respectively given by:*

$$ER^* = p_L + \pi \left(\frac{2\delta - 1}{\delta}\right)(p_H - p_L)$$

$$Eu^* = \pi(v_H - p_H) + (1 - \pi)(v_L - p_L) - (1 - \pi)\underline{\phi}(\delta, \pi, p_H)(p_H - p_L)$$

(ii) *A fraudulent hybrid H equilibrium is characterized by $\phi_H^* = 1$, $\phi_L^* = \left(\frac{\pi}{1-\pi}\right)\left(\frac{\delta}{1-\delta}\right)\left(\frac{v_H - p_H}{p_H - v_L}\right) :=$*
   *$\bar{\phi}(\delta, \pi, p_H), b^*(p_H, h) = \left(\frac{1}{1-\delta}\right)\frac{p_L}{p_H}$ and $b^*(p_H, l) = 0$.*
   *The ex-ante equilibrium payoffs of the seller and the buyer are respectively given by:*

$$ER^* = \left(1 + \pi\left(\frac{2\delta - 1}{1 - \delta}\right)\right)p_L; \quad Eu^* = (1 - \pi)(1 - \bar{\phi}(\delta))(v_L - p_L)$$

   *Furthermore, $b^*(p_H, l) = 1$ ($b^*(p_H, l) = 0$) in any Fraudulent Pooling L (Pooling H) equilibrium in which $\delta \notin \{\delta_l, \delta_h\}$.*

Similarly, the definition of the following variables will be proved useful for the rest of the paper.

**Definition 3.** *For any given any fraudulent PBE $(\mu^*, \phi^*, b^*)$, the equilibrium incidence of fraud is the consumer's ex-ante probability of becoming a victim of fraud:*

$$\Phi^* := (1 - \pi)\phi_L^*[(1 - \delta)b^*(p_H, h) + \delta b^*(p_H, l)].$$

*The ex-ante equilibrium level of trade is given by:*

$$t^* := [\pi\delta + (1 - \pi)(1 - \delta)\phi_L^*]b^*(p_H, h) + [\pi(1 - \delta) + (1 - \pi)\delta\phi_L^*]b^*(p_H, l) + (1 - \pi)(1 - \phi_L^*)$$

*The ex-ante equilibrium gains from trade are given by:*

$$TS^* := \pi[\delta b^*(p_H, h) + (1 - \delta)b^*(p_H, l)]v_H + (1 - \pi)[\phi_L^*((1 - \delta)b^*(p_H, h) + \delta b^*(p_H, l)) + (1 - \phi_L^*)]v_L$$

It is worth mentioning that with independence of the signal realization observed, trade always takes place between the buyer and the seller in what we call fraudulent pooling $L$ equilibrium (a fraudulent pooling equilibrium in which the marginal buyer is the one who observes the low signal). This fraudulent equilibrium can be supported for sufficiently noisy signals and sufficiently high priors (e.g., if $\delta < \delta_l$ and $p_H < \bar{v}$). Since the potential gains from trade are fully realized, any fraudulent pooling $L$ equilibrium is efficient. However, the incidence of fraud is at its highest possible value $(1 - \pi)$. Therefore, the realized gains from trade are not necessarily a good measure of the level of welfare in the marketplace as their distribution among participants is not accounted for.

## 6. The Value of Information

The purpose of this section is to analyse the impact of certain variables of interest on the (ex-ante) equilibrium level of fraud, incidence of fraud, trade, gains from trade and welfare.

### 6.1. Private Information

The non-monotonicity of the level of fraud on the precision of private information when the high price exceeds the ex-ante expected valuation of the good is due to a change in the identity of the marginal customer. For $p_H > \max\{\bar{v}, 2p_L\}$, a fraudulent hybrid H (hybrid L) outcome can be supported in equilibrium if and only if the customer's private information is sufficiently imprecise (precise). The marginal customer is the buyer who observes a high (low) signal in the fraudulent hybrid H (hybrid L) equilibrium. As the signal gets more precise, the marginal customer becomes more optimistic (pessimistic) about the expected quality of the good. Her higher (lower) willingness to pay induces the low-quality seller to aggravate (moderate) his dishonest behaviour in the fraudulent hybrid H (hybrid L) equilibrium outcomes. The low-quality seller's equilibrium pricing strategy is represented in Figure 2.

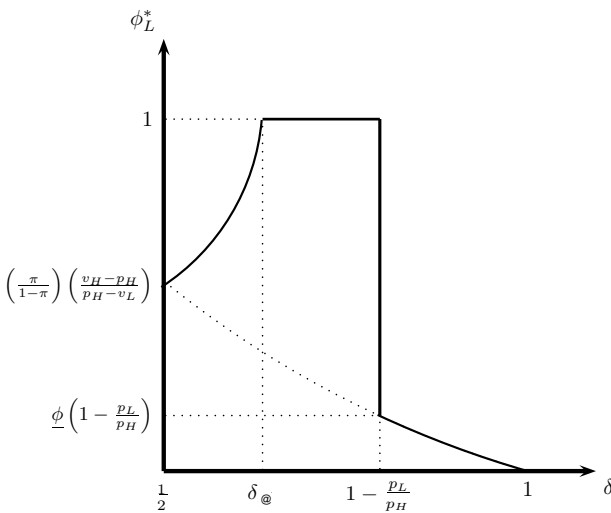

**Figure 2.** The equilibrium level of fraud as a function the signal precision $\delta$ if $p_H > \max\{\bar{v}, 2p_L\}$ and $\delta_h < 1 - (p_L/p_H)$.

**Proposition 2.** *The equilibrium level of fraud is non-increasing in the quality of the buyer's private information $\delta$ if $p_H \leq \max\{\bar{v}, 2p_L\}$. Instead, if $p_H > \max\{\bar{v}, 2p_L\}$, the equilibrium level of fraud is non-monotonic in the quality of the buyer's private information: more precision leads to a strictly higher level of fraud if $\delta < \min\{\delta_h, 1 - (p_L/p_H)\}$, and to a strictly lower level of fraud if $\delta \in (1 - (p_L/p_H), 1)$.*

Therefore, in any fraudulent hybrid H (hybrid L) equilibrium outcome, the more precise is the customer's private and exogenous information, the less (more) precise the high price as a public signal of quality is. This fact has further implications for trade, incidence of fraud and welfare.

**Corollary 2.** *Assume $p_H > \max\{\bar{v}, 2p_L\}$. The equilibrium incidence of fraud and the equilibrium level of trade are non-monotonic in the precision of private information.*

The impact of information provision on the variables of interest is as expected in fraudulent hybrid L equilibria: better private information provision increases the amount of information about quality revealed in the regulated market and it decreases the amount of fraud committed in the market. An implication of the complementary nature shared by both the "public" and private signals is that policies that favour accurate private information provision foster trade and lead to a Pareto improvement in any fraudulent hybrid L equilibrium. Indeed, total surplus is increasing and concave in the signal precision value.

We now proceed to discuss the effect of information provision in fraudulent hybrid H equilibria. The more precise the buyer's private information is, the less frequently the low-quality seller is

matched with the marginal customer who is the consumer with a high-signal realization. In order for the low-quality seller to remain indifferent between charging either price, the expected revenue associated with offering the good at the high price must remain constant and be equal to the low price. Thus, the lower probability of sale due to a less frequent match must be compensated with a higher acceptance rate of the high-price offer by the high-signal buyer when the match occurs. The higher level of fraud and acceptance rate by the high-signal buyer lead to a higher incidence of fraud and a higher equilibrium probability of trade for high-quality items in the fraudulent hybrid H equilibrium outcome (refer to Figure 3). Although a low-quality item which is offered at the low price is traded with certainty in the market, the more precise the customer's private information is, the less often low-quality goods are traded at the low price since the low-quality seller is more dishonest. In sum, the equilibrium ex-ante trade probability for low-quality units is decreasing in information ($t_L^* = 1 - \bar{\phi}(\delta, \pi, p_H)\,(1 - (p_L/p_H))$). The negative effect of information precision on trade of low quality products is outweighed by its positive effect on the trade of high-quality products, leading to a higher ex-ante expected level of trade, if and only if the low price is sufficiently high ($p_L > ((v_H - p_H)/(v_H - v_L))\,p_H$). This is because the higher the low price is, the higher the acceptance rate of high-quality products offered at the high-price must be for the low-quality seller to continue to be willing to randomize. Indeed, the equilibrium trading probability for both quality items is increasing in the low price. As a result, the gains from trade are increasing and convex in $\delta$ if and only if the low price is high enough ($p_L > ((v_H - p_H)/(v_H - v_L))\,v_L$). Otherwise, they are a decreasing and concave in the information precision:

$$TS^* = \pi \left(\frac{\delta}{1-\delta}\right) \frac{p_L}{p_H} v_H + (1-\pi) \left[1 - \bar{\phi}(\delta)\left(1 - \frac{p_L}{p_H}\right)\right] v_L < \bar{v}$$

Regarding welfare, the high-quality seller benefits from a more precise customer's information as it fosters the trade of his product. The low-quality seller's (expected) revenue must be constant and equal to the value of the low price for a randomization to be optimal. Therefore, the seller's ex-ante expected revenue is increasing in the signal accuracy. If $p_L < v_L$, only the buyer who is offered the item at the low-price can obtain a positive economic rent as the low-signal buyer optimally rejects a high price offer whereas the high-signal buyer is extracted her informational rent when receiving the high-price offer. Hence, if $p_L < v_L$, the buyer's ex-ante expected utility decreases with the level of signal precision due to the lower probability of offering the item at the low-price. Therefore, a higher precision of her private information makes the buyer become worse-off by the higher induced level of fraud.

**Corollary 3.** *Assume $p_H > \max\{\bar{v}, 2p_L\} > v_L > p_L$. A higher precision of private information reduces consumers' welfare if $\delta < \min\{\delta_h, 1 - (p_L/p_H)\}$.*

Consumer Protection Agencies are concerned about consumers' welfare and about protecting them from unfair commercial practices. As illustrated in the introduction, these agencies favour more accurate private information provision policies by the issue of institutional warnings. Our analysis suggests that there is room for a Consumer Protection Agency intervention either when the high price is sufficiently low or when it is sufficiently high as long as the information provided in the warnings is relatively precise in the latter case. When the high price is sufficiently high, issuing imprecise warnings or increasing the precision of the already relatively imprecise warnings may be detrimental for consumers by the generation of higher levels of fraud. Thus, a counter intuitive result of ours is the desirability of less intervention by Consumer Protection Agencies for some parameter values: issuing less (descriptive) warnings may be optimal sometimes. In such cases, consumer fraud could be alternatively alleviated via regulation (refer to Section 6.3).

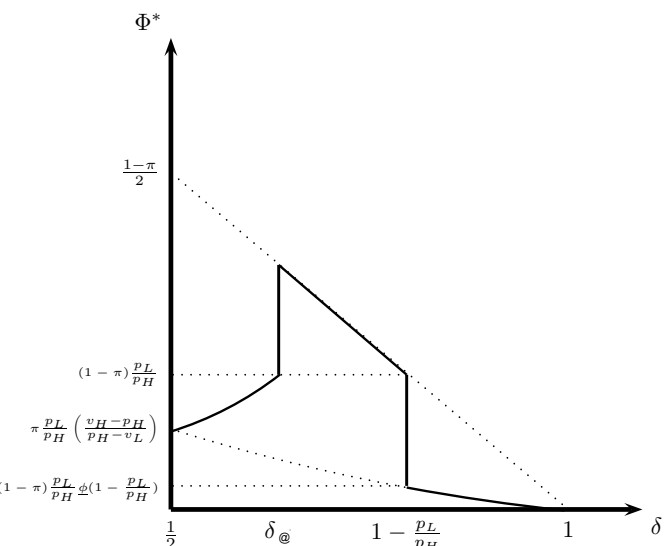

**Figure 3.** The equilibrium incidence of fraud as a function the signal precision $\delta$ if $p_H > \max\{\bar{v}, 2p_L\}$ and $\delta_h < 1 - (p_L/p_H)$.

Finally, it is worth discussing the pooling H equilibrium outcome in which the buyer who observes a high (low) signal realization purchases (does not purchase) the item offered at the high price with certainty. The reader is reminded that this fraudulent pooling H equilibrium outcome can be supported for intermediate signal precision values.[7] The more precise the customer's private signal is, the lower (higher) the probability that a low-quality (high-quality) seller is matched with a high-signal buyer is. As a result, the consumer's welfare is enhanced since the buyer with a favourable signal becomes more optimistic about the expected quality of the item on sale, increasing her informational rent ($Eu^* = \pi\delta(v_H - p_H) - (1 - \pi)(1 - \delta)(p_H - v_L)$).[8] Additionally, the equilibrium trade probabilities for high-quality and low-quality products are increasing and decreasing respectively ($t_H^* = \delta$ and $t_L^* = 1 - \delta$) in information, leading to a lower incidence of fraud in equilibrium ($\Phi^* = (1 - \pi)(1 - \delta)$). As a result, the ex-ante seller's expected revenue is increasing in the precision of information if and only if the prior is sufficiently high ($\pi > 1/2$) as the more precise the private signal, the higher (lower) the expected revenue by the high-quality (low-quality) seller is ($ER^* = [\pi\delta + (1 - \pi)(1 - \delta)]p_H$). Overall, total surplus is also increasing in information precision if and only if the prior is sufficiently high ($\pi > v_L/(v_H + v_L)$ as $TS^* = \pi\delta v_H + (1 - \pi)(1 - \delta)v_L < \bar{v}$). Therefore, for intermediate signal precision values, private information provision policies have no effect on the precision of the public information released strategically by the seller in equilibrium (as the price remains uninformative) but they lead socially to both a Pareto improvement in ex-ante terms and less fraud committed in the market if and only if the prior probability for high quality products is relatively high. Otherwise, these information provision policies will be detrimental for trade and suppliers but beneficial for consumers.

*6.2. Prior Belief*

By condition (2), the higher the prior is, the higher the consumer's posterior beliefs upon receiving a high price offer are, independently of the signal realization observed by the buyer (refer to Figure 2). The higher willingness to pay by the more optimistic consumer induces the low-quality seller to be more dishonest in fraudulent hybrid equilibria. Crucially, the consumer's probability of acceptance of a high-price offer does not depend on the prior in hybrid equilibria as this probability is pinned down

---

[7] Specifically, the range is given by $\delta \in (\delta_i, 1 - (p_L/p_H))$ where $\delta_i = \delta_h$ if $p_H \geq \bar{v}$ and $\delta_i = \delta_l$ if $p_H < \bar{v}$.
[8] Remember that the customer whose signal realization is low does not trade.

by the low-quality seller's price indifference condition. Therefore, the exacerbation of fraud increases the incidence of fraud and diminishes trade of low quality products.

The seller's ex-ante wellbeing is enhanced since the increase in the probability that the seller's item is of high-quality lowers his opportunity cost of signalling high quality. The overall impact of a higher prior on consumers' welfare is negative in fraudulent hybrid H equilibrium outcomes as its negative impact via an increase in the level of fraud and incidence of fraud dominates its positive effect by the increase in the ex-ante expected valuation of the good $\bar{v}$. Instead, the sign of its overall effect on consumers' welfare is ambiguous in hybrid L equilibrium outcomes since it depends on the level of precision of the private signal: it is positive if and only if the signal is sufficiently accurate, and negative otherwise. As a result, total surplus is linearly increasing (decreasing) in the prior in any fraudulent hybrid L equilibrium outcome if and only if the private information is sufficiently precise (imprecise) (e.g., $\delta > (<)(p_H - p_L)/(2p_H - v_L - p_L)$).

### 6.3. Regulation

A regulator can manipulate the prices of the products exogenously fixed in our market.

The value of the low price does not have an effect on the level of fraud in the market but it affects the incidence of fraud through two channels: a direct effect on the acceptance rate in fraudulent hybrid equilibria (as we discussed above) and an indirect effect on the support of a fraudulent hybrid equilibria. On the one hand, a higher low-price $p_L$ favours trade and hence, it increases total surplus in equilibrium but at a cost of a higher incidence of fraud. This has a negative (positive) impact on the ex-ante buyer's (seller's) expected utility. On the other hand, it favours the existence of fraudulent hybrid $L$ equilibrium vs. fraudulent pooling $H$, widening the range of signal precision values for which a fraudulent hybrid $L$ equilibrium outcome can be in supported. The change in the supported equilibrium type contributes to the reduction in the equilibrium incidence of fraud.

A higher high price $p_H$ disciplines the low-quality seller, lowering the incidence of fraud but aggravating the cost of fraud to the buyer in fraudulent hybrid equilibria.[9] The first effect dominates the second effect reducing the buyer's expected loss due to fraud in hybrid equilibria. An increase in the high-price leads to a Pareto superior hybrid H equilibrium outcome as only the buyer who is offered the low-quality product at the low price is able to extract some rent in this equilibrium. On the contrary, the buyer who observes the high signal is also able to extract some rent when the item is offered at the high price in a hybrid L equilibrium outcome, and therefore, we have two opposite effects on the buyer's expected rent. The overall effect is non-negative if and only if the signal is sufficiently imprecise, e.g., $\delta \leq \lambda/(1+\lambda)$ where $\lambda := [(v_H - v_L)(p_H - p_L) - (v_H - p_H)(p_H - v_L)]/(p_H - v_L)^2$. Finally, a higher high price $p_H$ favours the existence of fraudulent hybrid versus fraudulent pooling equilibria, inducing the low-quality seller to be more honest for a wider-range of signal precision values.

## 7. Discussion

This section briefly sketches how the main results of the model change when the information structure is modified to be more extreme, the private signal observed by the buyer becomes public and the consumer has the option of acquire perfect information at an expense.

### 7.1. Extreme Information Structures

In this subsection, we assume, as the vast majority of the papers in the literature do, that some consumers can ascertain the quality of the product perfectly by inspection whereas the remaining consumers are completely uninformed about the quality of the good and they ex-ante believe that

---

[9] Indeed, a higher high price fosters ex-ante trade in a fraudulent hybrid L equilibrium outcome but its effect on the ex-ante trade is ambiguous in a fraudulent hybrid H equilibrium outcome: it discourages trade if and only if the low price is sufficiently high, so that the negative effect on the trade of high-quality products dominates.

quality is high with probability $\pi$. Let $\alpha$ denote the probability that the consumer is perfectly informed. This case is closely related to the case in our model in which consumers are heterogeneous in the degree of signal precision: a fraction $\alpha$ of them observe a perfectly informative signal ($\delta_1 = 1$) whereas the rest observe an uninformative signal ($\delta_2 = 1/2$). The seller with the low quality product who charges the high price can potentially capture only an uninformed consumer since every informed consumer will accept the high price offer only if the seller's good is of high quality. Three main distinctions from the equilibrium results predicted by our model must be emphasized.

First, a sufficiently large fraction of informed consumers ($\alpha > 1 - (p_L/p_H)$) enables the high price to signal high quality, leading to the emergence of a separating equilibrium. By definition, there is no fraud committed in any separating equilibrium and the potential gains from trade are fully realized.

Second, if the high price is higher than the ex-ante expected valuation of the good ($p_H > \bar{v}$) and the fraction of informed consumers is not sufficiently large ($\alpha < 1 - (p_L/p_H)$), a unique hybrid equilibrium outcome exists: the high quality seller always charges the high price, the low quality seller randomizes strictly between setting both prices and charges the high price with a probability $\phi_L^* = \phi_u := (\pi/(1-\pi))((v_H - p_H)/(p_H - v_L))$, and the uninformed customer accepts the high price offer with probability $b_u^* = (1/(1-\alpha))(p_L/p_H)$.[10] On one hand, the cheating randomization strategy per se does not depend on the fraction of informed consumers (only its equilibrium support does) because the uninformed consumer's posterior beliefs do not depend on the fraction of informed consumers. On the other hand, the acceptance probability by the uninformed customer is increasing in the fraction of informed consumers. Ceteris paribus, the higher is this fraction, the higher is the probability that a low quality seller confronts an informed customer and therefore, the higher is the probability that a high price offer may end up being rejected. In order for this seller to remain indifferent and still be willing to randomize, the uninformed customer must accept the high price offer more frequently so that the probability of trade for the low quality seller remains invariant. Both facts imply that neither the incidence of fraud ($\Phi^* = \pi((v_H - p_H)/(p_H - v_L))(p_L/p_H)$) nor the ex-ante trade probability for low-quality items in equilibrium ($t_L^* = \phi_u(1-\alpha)b_u^* + (1-\phi_u) = 1 - \phi_u(1 - (p_L/p_H))$) depend on the fraction of informed consumers. Instead, the trading equilibrium probability for high-quality items does ($t_H^*(p_H) = \alpha + (1-\alpha)b_u^* = \alpha + (p_L/p_H)$) and it is increasing in this parameter since every informed consumer always accepts trading this item at the high price. As a result, a higher fraction of informed buyers in the market is beneficial for the seller ($ER^* = p_L + \alpha\pi p_H$) and for the realization of the potential gains from trade ($TS^* = \pi v_H(\alpha + (p_L/p_H)) + v_L[(1-\pi) - \pi((v_H - p_H)/(p_H - v_L))(1 - (p_L/p_H))]$). Finally, consumers always benefit (in ex-ante terms) from the presence of more informed consumers because these types of consumers never become victims of fraud ($Eu^* = \pi(v_H - p_H)(\alpha + (p_L/p_H)) + (1-\pi)(v_L - p_L) - \pi v_L((v_H - p_H)/(p_H - v_L))(1 - (p_L/p_H))$).

Third, if the high price is lower than the ex-ante expected valuation of the good ($p_H < \bar{v}$) and the fraction of informed consumers is not sufficiently large ($\alpha < 1 - (p_L/p_H)$), then a unique pooling equilibrium outcome exists in which both sellers charge the high price and every uninformed customer always accepts any offer. Trade is not monotonic in information because informed customers are never fooled into purchasing low-quality products at the high price. As a result, we have that: (i) the incidence of fraud is decreasing in the fraction of informed consumers; (ii) the ex-ante buyer's expected utility is increasing in the fraction of informed consumers; (iii) the seller's ex-ante expected utility and total surplus are decreasing in the fraction of informed consumers.

Contrary to the results from the original model, it is immediate that the equilibrium incidence of fraud is a non-increasing function of the fraction of informed consumers in the market (refer to Figure 4), and that consumers always benefit from more information in ex-ante expected terms.

---

[10] Clearly, both types of customer always accept the low price offer as in the original model.

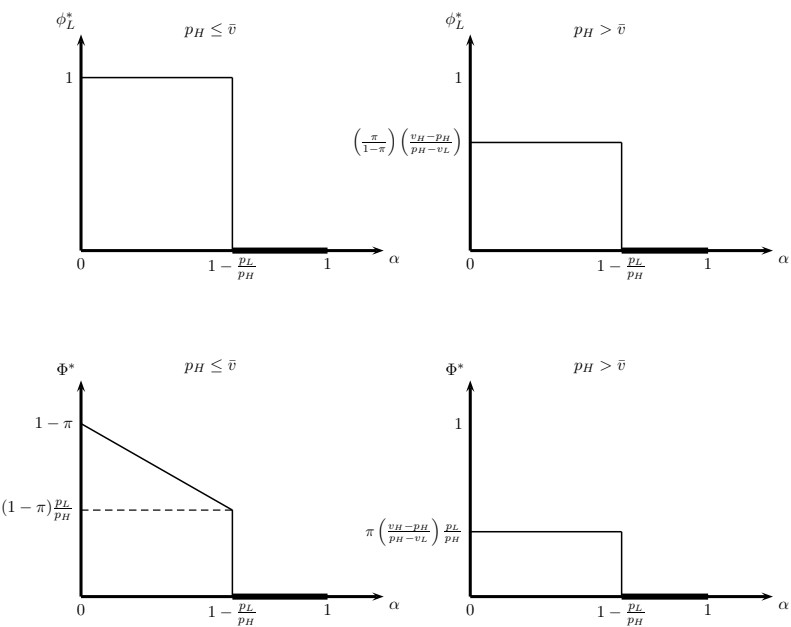

**Figure 4.** The equilibrium level of fraud and incidence of fraud as a function of the fraction of informed consumers $\alpha$.

Two additional main differences between both models are highlighted. First, since more information is always beneficial for all agents in expected terms under this extreme information structure if the high price is higher than the ex-ante expected valuation, the equilibria can be Pareto ranked according to the value of the fraction of informed consumers in the market. This result contradicts our previous result of a possible negative value of private information for some agents under this parameter configuration. Second, suppose that the high price is lower than the ex-ante expected valuation. Then, the effect of an increase in the fraction of informed consumers on the monopolist's ex-ante expected revenue and total surplus is negative under the extreme information structure if the fraction of informed consumers is sufficiently low ($\alpha < 1 - (p_L/p_H)$). Instead, this effect is positive (negative) for sufficiently high (low) prior beliefs if $\delta \in (\delta_I, 1 - (p_L/p_H))$ in the original model. The intuition is as follows. For these parameter configurations, the supported equilibrium is a pooling equilibrium in which only the fraction of buyers with the favourable signal (who are uninformed) purchase the low quality item at the high price under the original (extreme) information structure. The critical difference between both settings is that the fraction of uninformed consumers is exogenously fixed, it does not depend on the state, under the extreme information structure whereas the fraction of consumers who receive a favourable signal depends on the state and therefore, it is positively correlated with value of the prior belief in the original model.

### 7.2. Public versus Private Information

This subsection briefly sketches how the main results of the model change when the consumer's private imperfect information about the good's quality becomes public. The model then becomes a one-sided asymmetric information model. Because market prices are fixed, they cannot be made contingent on the realization of the public signal. As in [17], the seller replaces price discrimination with cheating his identifiable customers selectively.

The main difference with respect to our previous analysis is that the ex-post opportunity cost of signalling high quality by charging the high price (in terms of the trade probability) is the same for both types of seller. As a result, for each value of the signal precision, there exists a pooling equilibrium in

which the prices offered are not signals of quality.[11] Let $\phi^s$ denote the probability of offering the item at the high price by both seller types once signal $s$ is publicly observed. Three cases can be distinguished for generic values of $\delta$.

First, if the high price is relatively high (e.g., if $p_H > \bar{v}$ and $\delta < \delta_h$),[12] the sellers' optimal strategy is to always pool on the low price so that there is no fraud committed in equilibrium. The inability of the seller with a high-quality item to signal its quality when the precision of public information is relatively low forces him to offer the item at the low-price in order to make positive profits. The equilibrium is efficient as the potential gains from trade are fully realized. A fraudulent hybrid H equilibrium outcome was supported in the two-sided information asymmetric model under this parameter configuration. As there is no fraud committed in equilibrium when the signal is made public, the buyer is unambiguously strictly better off in the one-sided asymmetric information model relative to the two-sided asymmetric information model. Instead, the revelation of the buyer's private signal is beneficial for the seller if and only if the high price is not too high so that the ex-ante probability of confronting a low-signal buyer, who would reject the high price offer in the fraudulent hybrid H, is sufficiently high ($\pi(1 - \delta) + (1 - \pi)\delta > 1 - (p_L/p_H)$).

Second, if the high price is relatively low (e.g., if $p_H < \bar{v}$ and $\delta < \delta_l$),[13] the sellers' optimal strategy is to always pool on the high price and the buyer always agrees to trade at this price independently of the signal realization observed. Therefore, the same equilibrium outcome (fraudulent pooling L equilibrium) is supported under the one-sided and two-sided asymmetric information models. The incidence of fraud is at its highest possible value in equilibrium. The potential gains from trade are fully realized so the equilibrium is efficient. The seller's ex-ante expected revenue and buyer's ex-ante expected utility do not depend on the precision of information in this region.

Third, if the high price is of intermediate value, or equivalently, if the public information is sufficiently precise (e.g., if $p_H > \bar{v}$ and $\delta > \delta_h$ or if $p_H < \bar{v}$ and $\delta > \delta_l$), the strategy of both seller types optimally depends on the signal realization publicly observed: the item is offered at the high (low) price in equilibrium if a high-signal (low-signal) realization is publicly observed ($\phi^h = 1 > \phi^l = 0$). Trade is always accepted in equilibrium and the expected gains of trade are fully realized so that equilibrium is efficient. As public information becomes more precise, the observation of a high signal becomes more rare if the good on sale is of low-quality. Consequently, the incidence of fraud is lower ($\Phi^* = (1 - \pi)(1 - \delta)$) and in the limit, it approaches a zero value as the signal becomes arbitrarily precise. The seller's and buyer's ex-ante expected utilities are respectively given by:

$$ER^* = p_L + [\pi\delta + (1 - \pi)(1 - \delta)](p_H - p_L)$$

$$Eu^* = \pi(v_H - p_H) + (1 - \pi)(v_L - p_L) + (2\pi - 1)(1 - \delta)(p_H - p_L)$$

The value of public information is negative to the buyer (seller) if and only if $\pi > (<)1/2$.

For the parameter configuration that supported a fraudulent pooling H equilibrium under the two-sided asymmetric information model ($p_H > \bar{v}$ and $\delta \in (\delta_h, 1 - (p_L/p_H))$ or $p_H < \bar{v}$ and $\delta \in (\delta_l, 1 - (p_L/p_H))$), the public revelation of the buyer's private information is a Pareto improvement since whenever a buyer observes a low signal there is trade in the market as the item is offered at the low price instead of the high price. Finally, for the parameter configuration that supported a

---

[11] There also exists a continuum of hybrid equilibria outcomes in accordance with the sellers' randomization strategies which are themselves not pinned down but their ratio is pinned down. For example, any strategies $\bigcup_{\theta \in \Theta, s \in S} \{\phi_\theta^{*s}\}$ and $\bigcup_{s \in S} \{b^*(p_H, s), b^*(p_L, s)\}$ such that $\phi_L^{*h}/\phi_H^{*h} = \bar{\phi}(\delta, \pi, p_H)$, $\phi_L^{*l}/\phi_H^{*l} = \underline{\phi}(\delta, \pi, p_H)$, $b^*(p_H, h) = b^*(p_H, l) = p_L/p_H$ and $b^*(p_L, h) = b^*(p_L, l) = 1$ can be supported in equilibrium. Unfortunately, given that the equilibrium dishonesty level is not pinned down, comparative statics cannot be performed.

[12] Notice that the high-signal buyer's indifference threshold $\delta_h$ is increasing in the high price $p_H$ and thus, for any given signal precision value, these conditions are simultaneously satisfied if the high price is sufficiently high.

[13] Notice that the low-signal buyer's indifference threshold $\delta_l$ is decreasing in the high price $p_H$ and thus, for any given signal precision value, these conditions are simultaneously satisfied if the high price is sufficiently low.

fraudulent hybrid L equilibrium under the two-sided asymmetric information model ($p_H > \bar{v}$ and $\delta > 1 - (p_L/p_H)$ or $p_H < \bar{v}$ and $\delta > max\{\delta_l, 1 - (p_L/p_H)\}$), the seller is strictly better off in the one-sided versus two-sided asymmetric information model as he can now tailor his cheating behaviour to the identifiable consumer's characteristic (her observable signal). On one hand, the purchase of high-quality items at the low price when the low signal is observed benefits the buyer in the one-sided asymmetric information model relative to the two-sided asymmetric information model. On the other hand, the outright cheating behaviour (instead of partial fraudulent behavior) by the low-quality seller when the high-signal is observed worsen the buyer in the one-sided asymmetric information relative to the two-sided asymmetric information model. Overall, the buyer benefits from the public revelation of her private information if and only if either the prior is greater than one half ($\pi \geq 1/2$) or if it is lower than one half and information is sufficiently imprecise ($\pi < 1/2$ and $\delta < (\pi/(1 - 2\pi))\left((v_H - p_H)/(p_H - v_L)\right)$). The incidence of fraud is weakly lower when the signal is revealed publicly than when it remains private if the high price is relatively high ($p_H > \max\{\bar{v}, 2p_L\}$). The opposite result occurs if the high price is lower than the ex-ante valuation of the good ($p_H < \bar{v}$) and the information is sufficiently precise.[14]

The non-monotonicity of fraud in information is robust when information is made public. If the high price is higher than the ex-ante valuation of the good, we already showed that the equilibrium is truthful when the public information is not sufficiently precise whereas the low-quality seller cheats selectively when the public information is sufficiently precise. As a result, there is a discontinuity in the level of fraud in the market: the incidence of fraud is zero for low signal precision values whereas it is strictly positive for high signal precision values although its value decreases with the precision of public information.

In sum, there are three main differences in terms of equilibrium outcomes when comparing a one-sided versus a two-sided asymmetric information model. First, the public revelation of the consumer's information favours the existence of a truthful equilibrium. Contrary to our previous results, fraud can now be completely eradicated by establishing a sufficiently high price in the one-sided asymmetric information model. Second, it is immediate that total surplus is unambiguously higher when consumers' private information becomes public because the potential gains from trade are fully realized for all parameter values in the one-sided asymmetric information model. Third, if the high price is either sufficiently high ($p_H > \max\{\bar{v}, 2p_L\}$) or sufficiently low ($p_H < \bar{v}$ and $\delta_l > 1 - (p_L/p_H)$), the value of public (private) information can be found negative for some agent, the buyer or the seller depending on the value of the prior, only if information is (not) sufficiently precise.

### 7.3. Costly Perfect Information Acquisition

In the spirit of [3] (work), before making the purchasing decision, we now allow consumers to decide whether or not to acquire perfect information about quality at a cost $c > 0$ once the price offer and the noisy signal realization have been observed. This test is not publicly observable and its outcome cannot be credibly communicated. Note that consumers may acquire information only if the item is offered at the high price as the low price falls short of her valuation for any item. The acquisition of perfect information may also be conditioned on the signal realization observed. Once perfect information is acquired, consumers would only buy the item at the high price if the perfectly precise signal reveals high quality. As a result, a low quality seller who sets a high-price offer may only sell his product to those consumers who do not invest in information acquisition. Interestingly, consumers' outside option of no information acquisition depends on the accuracy of their freely noisy signal. Therefore, a consumer is willing to pay for perfect information if and only if it is not too costly: $c \leq \min\{(1 - \mu(H|p_H, s; \delta, \phi_L))(p_H - v_L), \mu(H|p_H, s; \delta, \phi_L)(v_H - p_H)\}$.

---

[14] Specifically, $\delta > \left[1 + [(\pi/(1 - \pi))((v_H - p_H)/(p_H - v_L))(p_L/p_H)]^{-1}\right]^{-1}$.

If the cost of information acquisition is relatively high, $c > (v_H - p_H)(p_H - v_L)/(v_H - v_L) := \bar{c}$, no consumer has incentives to acquire perfect information and the same hybrid equilibria of the original model remain valid. We focus on the case in which this cost is low, $c \leq \bar{c}$, so that some consumers invest in information in equilibrium. As in [3], there is no separating equilibrium: if it were, consumers would not have incentives to acquire perfect information since its acquisition is costly and prices already revealed quality perfectly. However, if no consumer invested in information, only pooling or partial-pooling strategies can be supported in equilibrium. Therefore, prices cannot be perfectly informative in equilibrium, as in our original model. Contrary to [3], pooling on the high price can be supported in equilibrium. With price rigidities, the high-quality seller cannot separate himself by charging higher prices in order to capture those who invested in information acquisition. Hence, prices can be completely non-informative as in the original model. However, capturing both consumer types by pooling on the high price (Pooling L) is no longer an equilibrium outcome if perfect information is inexpensive since the existence of Pooling L requires no consumer to invest in perfect information acquisition. Those Pooling L equilibrium outcomes that could be supported in the original model become now partial-pooling (H) equilibria in the new model.

The access to inexpensive perfect information leads to regions of the parameter space that support the existence of partial pooling equilibrium outcomes in which the low-quality seller quotes the high price with a strictly positive probability lower than one. In the partial-pooling H equilibrium outcome, the low-quality seller imitates the high-quality seller with probability $(\pi/(1-\pi))(\delta/(1-\delta))(c/(p_H - v_L - c))$ and one of the following holds: either (i) the consumer who observes the low-signal realization optimally invests in information acquisition whereas the consumer who observes the high-signal realization randomizes, not acquiring the perfect signal sometimes, and she optimally purchases the good offered at the high price whenever no investment is made in information; or (ii) the consumer who observes the low-signal realization randomizes in information acquisition and she does not purchase the good whenever she does not become informed whereas the consumer who observes the high-signal realization optimally does not perform the test and always purchases the good offered at the high price. In the partial-pooling L equilibrium outcome, the low-quality seller imitates the high-quality seller with probability $(\pi/(1-\pi))((1-\delta)/\delta)(c/(p_H - v_L - c))$, the consumer who observes the low-signal realization randomizes in information acquisition and she does purchase the good whenever she does not invest in information whereas the consumer who observes the high-signal realization optimally does not perform the test and always purchases the good offered at the high price. Interestingly, both partial-pooling equilibrium outcomes approach the full information outcome as the cost of information acquisition becomes negligible. When the cost of a perfect test is small, the information revealed by prices remains noisy in partial-pooling equilibrium outcomes making the non-monotonic result of fraud in information robust: the higher the precision of the free but noisy private information is, the higher the probability that the low-quality seller quotes the high price in the partial-pooling H equilibrium outcome is, and the more prevalent the level of fraud in the marketplace is.

## 8. Conclusions

This article has explored the role played by the accuracy of consumers' private information on the equilibrium levels of fraud, incidence of fraud and welfare in regulated markets with monopoly power and two-sided information asymmetries. In brief, our main findings are three-fold. First, equilibria involving fraud exist for all parameter values as prices reveal private information only imperfectly. Second, the level of fraud and incidence of fraud are not monotonic in information for some parameter values. Third, a more precise information can harm the buyer and/or the seller for some parameter values. We provided conditions on the parameters for the public revelation of the buyer's private information to be a Pareto improvement.

The model could be extended in several directions. Introducing competition among sellers and checking the robustness of these results in a oligopolistic market is deemed natural. Ref. [20] pursues

such analysis in a dynamic setting. Two key features of the model are that: (i) competition exists only on the sellers' side of the market; and that (ii) the model pertains to markets in which stores which are successful in selling their merchandise in a given period are more likely to receive the visit of a customer in the following period. The analysis shows that our main findings are robust to the introduction of competition among sellers. Similarly, incorporating reputation concerns among sellers into the model or modelling the endogenous acquisition of costly and noisy information are additional directions for future research.

**Funding:** Financial support from the Ministerio de Ciencia e Innovación and FEDER funds under project SEJ2007-62656 is gratefully acknowledged.

**Acknowledgments:** I thank two anonymous referees and seminar participants at the University of Toronto and University of Alicante for their comments. Specifically, I benefited from genuinely fruitful discussions with Ettore Damiano, Martin J. Osborne, Aloysius Siow, Sumon Majumdar and Miguel Sánchez Villalba. This work is based on a chapter of my Ph.D. dissertation (Ref. [20]). An earlier version of this work circulated as Instituto Valenciano de Investigaciones Económicas WP-AD-2014-01.

**Conflicts of Interest:** The author declares no conflict of interest.

## Appendix A

**Proof of Lemma 1.** We already proved in the text that $\phi_H^* \geq \phi_L^*$. It only remains to prove that no separating equilibrium in which the high type seller sets the high price and the low type seller sets the low price nor hybrid equilibria in which the high type seller strictly randomizes exist for any $\delta \in (1/2, 1)$. The proof is by contradiction.

(i) Suppose that a separating equilibrium, in which the high type seller sets the high price and the low type seller sets the low price, exists. Consider an information set $(p_H, s) \forall s \in S$ of the buyer. By consistency of beliefs along the equilibrium path, she believes that the item is of high quality with probability one. Because $v_H > p_H$, her optimal decision is to purchase the product for all possible signal realizations. However, this implies that the low quality seller could increase his expected payoff by charging the high price. A contradiction.

(ii) Suppose that an hybrid equilibrium in which the high type seller strictly randomizes between charging both prices and the low quality seller charges only the low price, exists. This results in a contradiction by the same argument as in (i).

(iii) Suppose that an hybrid equilibrium, in which both seller types strictly randomize between charging both prices, exists. Then, the low quality seller must be indifferent between charging either price. However, the high quality seller's strictly higher probability of trade at this price, relative to that of the low quality seller, implies that the high quality seller is not optimizing: he could increase his expected payoff by charging exclusively the high price. A contradiction.

□

**Proof of Proposition 1.** We prove this proposition through a series of lemmas.

**Lemma A1.** *Assume $p_H \leq \bar{v}$. Fraudulent Pooling equilibria exist if and only if $\frac{1}{2} < \delta \leq \max\left\{\delta_l, 1 - \frac{p_L}{p_H}\right\}$*

**Proof.** For necessity, suppose not, so that if $p_H \leq \bar{v}$, fraudulent pooling equilibria exist for some $\delta > \max\{\delta_l, 1 - (p_L/p_H)\}$. The buyer's posterior beliefs $\mu(H|p_H, s; 1)$ must be given by Equation (2) for all $s \in S$ by Bayes' consistency. If $\delta > \delta_l$, then $\mu(H|p_H, h; 1) > \pi \geq (p_H - v_L)/(v_H - v_L) > \mu(H|p_H, l; 1)$ so that the buyer's optimal strategy conditional on the signal received is given by $b^*(p_H, h) = 1$ and $b^*(p_H, l) = 0$. The low quality seller's expected revenue associated with charging the high price $p_H$ is given by $(1 - \delta)p_H < p_L$ due to the fact that $\delta > 1 - (p_L/p_H)$. The low quality seller would find it more profitable to deviate and charge exclusively the low price. A contradiction. We prove sufficiency through Characterizations A1–A3.

**Characterization A1.** *Assume $p_H \leq \bar{v}$ and $\delta < \delta_l$. There exists a Fraudulent Pooling equilibrium outcome characterized by the following strategies $\phi_H^* = \phi_L^* = 1$ and $b^*(p_H, h) = b^*(p_H, l) = 1$. There does not exist any other fraudulent equilibrium outcome.*

**Proof.** Consider any buyer's posterior beliefs $\mu^*$ such that $\mu^*(H|p_H, s; 1)$ is given by Equation (2) for all $s \in S$. To verify that $(\mu^*, \phi^*, b^*)$ is a PBE, note that these beliefs are by construction consistent with Bayes' rule along the equilibrium path. If $\delta < \delta_l$ then $\mu^*(H|p_H, h; 1) > \pi > \mu^*(H|p_H, l; 1) > (p_H - v_L)/(v_H - v_L)$ by definition of $\delta_l$ so that the buyer's strategy is rational by (1). Because the buyer agrees to trade at both prices always, and $p_H > p_L$, charging the high price is the best reply for both types of seller. Uniqueness follows trivially. □

**Characterization A2.** *Assume $p_H \leq \bar{v}$ and $\delta = \delta_l$. There exists a continuum of Fraudulent Pooling equilibrium outcomes characterized by $\phi_H^* = \phi_L^* = 1$, $b^*(p_H, h) = 1$ and $b^*(p_H, l) \in [1 - (1/\delta)(1 - (p_L/p_H)), 1]$.*

**Proof.** Consider any buyer's posterior beliefs $\mu^*$ such that $\mu^*(H|p_H, s; 1)$ given by Equation (2) for all $s \in S$. To verify that $(\mu^*, \phi^*, b^*)$ is a PBE, note that these beliefs are by construction consistent with Bayes' rule along the equilibrium path. If $\delta = \delta_l$, then $\mu^*(H|p_H, h; 1) > (p_H - v_L)/(v_H - v_L) = \mu^*(H|p_H, l; 1)$. The buyer's strategy is rational by (1). Hence, if the low quality seller posts the high price, his expected revenue is given by $[(1 - \delta) + \delta b^*(p_H, l)]p_H \geq p_L$. Thus, charging the high price is a best reply for both types of seller. □

**Characterization A3.** *Assume $p_H \leq \bar{v}$ and $\delta \in (\delta_l, 1 - (p_L/p_H)]$. There exists a Fraudulent Pooling equilibrium outcome characterized by $\phi_H^* = \phi_L^* = 1$, $b^*(p_H, h) = 1$ and $b^*(p_H, l) = 0$. There does not exist any other fraudulent equilibrium outcome.*

**Proof.** Consider any buyer's posterior beliefs $\mu^*$ such that $\mu^*(H|p_H, s; 1)$ is given by Equation (2) for all $s \in S$. To verify that $(\mu^*, \phi^*, b^*)$ is a PBE, note that these beliefs are by construction consistent with Bayes' rule along the equilibrium path. If $\delta > \delta_l$, then $\mu^*(H|p_H, h; 1) > (p_H - v_L)/(v_H - v_L) > \mu^*(H|p_H, l; 1)$ and the buyer's strategy is rational by (1). Because the buyer agrees to trade at the high price only if she observes the high signal realization, the low-quality seller's expected revenue if he charges the high price $p_H$ is given by $(1 - \delta)p_H \geq p_L$. Hence, charging the high price is a best reply by both seller types. Uniqueness follows from Lemma A2. □

This completes the proof of Lemma A1. □

**Lemma A2.** *Assume $p_H \leq \bar{v}$. Fraudulent Hybrid Equilibria exist if and only if $\delta \geq \max\{\delta_l, 1 - (p_L/p_H)\}$.*

**Proof.** For necessity, suppose not, so that if $p_H \leq \bar{v}$, hybrid equilibria exist for some $\delta < \max\{\delta_l, 1 - (p_L/p_H)\}$. Consider any buyer's posterior beliefs $\mu^*$ such that $\mu^*(H|p_H, s; \phi_L^*)$ are given by Equation (2) for all $s \in S$, satisfying Bayes' consistency. For any $p_H \leq \bar{v}$ and $\delta > 1/2$, $\mu^*(H|p_H, h; \phi_L^*) > (p_H - v_L)/(v_H - v_L) \; \forall \phi_L^* \in (0, 1)$, implying $b^*(p_H, h) = 1$. This strategy and $\delta < 1 - (p_L/p_H)$ imply that the low quality seller's expected revenue associated with charging the high price is $(1 - \delta)p_H > p_L$. Consequently, the low quality seller would find it more profitable to deviate and charge the high price rather than randomize. For sufficiency, consider the equilibrium outcome characterized by $\phi_H^* = 1$, $\phi_L^* = \phi(\delta, \pi, p_H)$, $b^*(p_H, h) = 1$ and $b^*(p_H, l) = 1 - (1/\delta)(1 - (p_L/p_H))$. Let the buyer's posterior beliefs be such that $\mu^*(H|p_H, s; \phi_L^*)$ is given by Equation (2) for all $s \in S$, satisfying Bayes' consistency along the equilibrium path. To verify that $(\mu^*, \phi^*, b^*)$ is a PBE if $\delta \geq \max\{\delta_l, 1 - (p_L/p_H)\}$, note that $\delta \geq \delta_l$ implies $\mu^*(H|p_H, h; \phi_L^*) > (p_H - v_L)/(v_H - v_L) = \mu^*(H|p_H, l; \phi_L^*)$ so that the buyer's strategy is rational by (1). Because the buyer accepts to trade at the high price with probability one if she observes the high signal realization and with probability

strictly less than one if she observes the low signal, the low quality seller's expected revenue if he charges the high price is given by $[(1 - \delta) + \delta b^*(p_H, l)]p_H = p_L$. The low quality seller is indifferent between charging either price and therefore, a randomization strategy is a best reply for this seller. Charging the high price is the high quality seller's only optimal response given the buyer's strategy. If $\delta > \max \{\delta_l, 1 - (p_L/p_H)\}$, this equilibrium outcome is the unique fraudulent equilibrium outcome by Lemma A1 and the fact that $b^*(p_H, h) = 1$ for all $\phi_L^* \in (0, 1)$ given any $p_H \leq \bar{v}$ and $\delta > 1/2$. This completes the proof of Lemma A2. $\square$

**Lemma A3.** *Assume* $p_H > \bar{v}$ *and* $\delta_h \leq 1 - (p_L/p_H)$. *Fraudulent Pooling equilibria exist if and only if* $\delta \in [\delta_h, 1 - (p_L/p_H)]$.

**Proof.** For necessity, suppose not, so that if $p_H > \bar{v}$ and $\delta_h \leq 1 - (p_L/p_H)$, fraudulent pooling equilibria exist for some $\delta \in (1/2, \delta_h) \cup (1 - (p_L/p_H), 1)$. By Bayes' consistency, the posterior beliefs $\mu^*(H|p_H, s; 1)$ must be given by Equation (2) for all $s \in S$. If $\delta < \delta_h$, then $(p_H - v_L)/(v_H - v_L) > \mu^*(H|p_H, h; 1) > \mu^*(H|p_H, l; 1)$ so that the buyer's optimal response to $\phi_L^* = 1$ is $b^*(p_H, h) = b^*(p_H, l) = 0$. Any seller who charged the high price would see his offer rejected and therefore, a low quality seller would be strictly better off if he charges the low price. If $\delta \in (1 - (p_L/p_H), 1)$, $\mu^*(H|p_H, h; 1) > (p_H - v_L)/(v_H - v_L) > \mu^*(H|p_H, l; 1)$, so that the buyer's optimal strategy is given by $b^*(p_H, h) = 1$ and $b^*(p_H; l) = 0$. The low quality seller's expected revenue associated with charging the high price is given by $(1 - \delta)p_H < p_L$ since $\delta > 1 - (p_L/p_H)$. The low quality seller would find it more profitable to deviate and charge the low price. A contradiction. We prove sufficiency through Characterizations A4–A5:

**Characterization A4.** *Assume* $p_H > \bar{v}$ *and* $\delta = \delta_h$. *There exists a continuum of pooling equilibrium outcomes characterized by* $\phi_H^* = \phi_L^* = 1$, $b^*(p_H, h) \in [(1/(1 - \delta))(p_L/p_H), 1]$ *and* $b^*(p_H, l) = 0$.

**Proof.** Consider any buyer's posterior beliefs $\mu^*$ such that $\mu^*(H|p_H, s; 1)$ is given by Equation (2) for all $s \in S$, satisfying Bayes' consistency along the equilibrium path. To verify that $(\mu^*, \phi^*, b^*)$ is a PBE, note that $\mu^*(H|p_H, h; 1) = (p_H - v_L)/(v_H - v_L) > \mu^*(H|p_H, L; 1)$ if $\delta = \delta_h$ implying that the buyer's strategy is rational by (1). The low quality seller's expected revenue conditional on charging the high price is given by $(1 - \delta)b^*(p_H, h)p_H \geq p_L$. Thus, charging the high price is a best reply for both seller types. $\square$

**Characterization A5.** *Assume* $p_H > \bar{v}$ *and* $\delta \in (\delta_h, 1 - (p_L/p_H)]$. *There exists a pooling equilibrium outcome characterized by* $\phi_H^* = \phi_L^* = 1$, $b^*(p_H, h) = 1$ *and* $b^*(p_H, l) = 0$. *This equilibrium outcome is the unique fraudulent equilibrium outcome if* $\delta < 1 - (p_L/p_H)$.

**Proof.** Consider any buyer's posterior beliefs $\mu^*$ such that $\mu^*(H|p_H, s; 1)$ is given by Equation (2), satisfying Bayes' rule along the equilibrium path by construction. To verify that $(\mu^*, \phi^*, b^*)$ is a PBE, note that $\mu^*(H|p_H, h; 1) > (p_H - v_L)/(v_H - v_L) > \mu^*(H|p_H, l; 1)$ for any $\delta > \delta_h$. Hence, the buyer's strategy is rational by (1). Since the buyer agrees to trade at the high price only if she observes the high signal realization, the low quality seller's expected revenue associated with charging the high price is given by $(1 - \delta)p_H \geq p_L$. As a result, charging the high price is a best reply for both types of the seller. Uniqueness of this fraudulent equilibrium outcome follows from Lemma A4. $\square$

This completes the proof of Lemma A3. $\square$

**Lemma A4.** *Assume* $p_H > \bar{v}$ *and* $\delta_h \leq 1 - (p_L/p_H)$. *Fraudulent Hybrid Equilibria exist if and only if* $\delta \in (1/2, \delta_h) \cup [1 - (p_L/p_H), 1)$.

**Proof.** For necessity, suppose not, so that if $p_H > \bar{v}$ and $\delta_h \leq 1 - (p_L/p_H)$, fraudulent hybrid equilibria exist if $\delta \in [\delta_h, 1 - (p_L/p_H))$. By Bayes' consistency along the equilibrium path, the buyer's

posterior beliefs $\mu^*(H|p_H, s; \phi_L^*)$ must be given by Equation (2) for all $s \in S$. Then $\mu^*(H|p_H, h; \phi_L^*) > (p_H - v_L)/(v_H - v_L)$ for all $\phi_L^* \in (0,1)$ and $\delta \geq \delta_h$ implying $b^*(p_H, h) = 1$ for all $\phi_L^* \in (0,1)$ and $\delta \geq \delta_h$. The low quality seller's expected revenue if he charges the high price is given at least by $(1 - \delta)p_H > p_L$, where the inequality follows from $\delta < 1 - (p_L/p_H)$. Hence, the low quality seller would find it more profitable to deviate and charge the high price instead of randomizing. A contradiction. We prove sufficiency through Characterizations A.4.1–A.4.2. Note that there do not exist other fraudulent outcomes for $\delta \in (1/2, \delta_h) \cup (1 - (p_L/p_H), 1)$ by Lemma A3.

**Characterization A6.** *Assume $p_H > \bar{v}$ and $\delta < \min\{\delta_h, 1 - (p_L/p_H)\}$. There exists a Fraudulent Hybrid Equilibrium outcome characterized by $\phi_H^* = 1$, $\phi_L^* = \bar{\phi}(\delta, \pi, p_H)$, $b^*(p_H, h) = (1/(1-\delta))\,(p_L/p_H)$ and $b^*(p_H, l) = 0$.*

**Proof.** Consider any buyer's posterior beliefs $\mu^*$ such that $\mu^*(H|p_H, s; \phi_L^*)$ is given by Equation (2) for all $s \in S$. To verify that $(\mu^*, \phi^*, b^*)$ is a PBE, note that $\mu^*(H|p_H, h; \bar{\phi}(\delta, \pi, p_H)) = (p_H - v_L)/(v_H - v_L) > \mu^*(H|p_H, l; \bar{\phi}(\delta, \pi, p_H))$ if $\delta < \delta_h$ implying that the buyer's strategy is rational by (1). Given the buyer's strategy, the low quality seller's expected revenue if he charges the high price is given by $(1 - \delta)b^*(p_H, h)p_H = p_L$ so that a randomization strategy is a best reply for the low-quality seller. Charging exclusively the high price is an optimal response by the high-quality seller's to the buyer's strategy. $\square$

**Characterization A7.** *Assume $p_H > \bar{v}$ and $\delta \geq \max\{\delta_h, 1 - (p_L/p_H)\}$. There exists a Fraudulent Hybrid Equilibrium outcome characterized by $\phi_H^* = 1$, $\phi_L^* = \underline{\phi}(\delta, \pi, p_H)$, $b^*(p_H, h) = 1$ and $b^*(p_H, l) = 1 - (1/\delta)\,(1 - (p_L/p_H))$.*

**Proof.** Consider any buyer's posterior beliefs $\mu^*$ such that $\mu^*(H \mid p_H, s; \phi_L^*)$ is given by Equation (2) for all $s \in S$ satisfying consistency of beliefs along the equilibrium path. To verify that $(\mu^*, \phi^*, b^*)$ is a PBE, note that $\mu^*(H|p_H, h; \underline{\phi}(\delta, \pi, p_H)) > (p_H - v_L)/(v_H - v_L) = \mu^*(H|p_H, l; \underline{\phi}(\delta, \pi, p_H))$ if $\delta \geq \delta_h$. Therefore, the buyer's strategy is rational by (1). Given the buyer's strategy, the low quality seller's expected revenue associated with the high price is given by $[(1-\delta) + \delta b^*(p_H, l)]p_H = p_L$. The low quality seller is indifferent between charging either price and therefore, a randomization strategy is a best reply for this seller. Thus, charging the high price is an optimal response by the high quality seller to the buyer's strategy. $\square$

This completes the proof of Lemma A4. $\square$

**Lemma A5.** *Assume $p_H > \bar{v}$ and $\delta_h > 1 - (p_L/p_H)$. No Fraudulent Pooling equilibrium exists whereas Fraudulent Hybrid equilibria exist for all $\delta \in (1/2, 1)$.*

**Proof.** Suppose that a fraudulent pooling equilibrium exists for some $\delta \in (1/2, 1)$ if $p_H > \bar{v}$ and $\delta_h > 1 - (p_L/p_H)$. By Bayes consistency, the posterior beliefs $\mu^*(H|p_H, s; 1)$ must be given by Equation (2) for all $s \in S$. If $\delta < \delta_h$, then $(p_H - v_L)/(v_H - v_L) > \mu^*(H|p_H, h; 1) > \mu^*(H|p_H, l; 1)$ so that the buyer's optimal strategies are $b^*(p_H, h) = b^*(p_H, l) = 0$. Any seller who charged the high price would see his price offer rejected. Thus, a low quality seller would be strictly better off deviating and charging the low price. If $\delta \geq \delta_h$, $\mu^*(H|p_H, h; 1) \geq (p_H - v_L)/(v_H - v_L) > \mu^*(H|p_H, l; 1)$, so that the buyer's optimal strategies are $b^*(p_H, h) \leq 1$ and $b^*(p_H, l) = 0$. The low quality seller's expected profit if he charges the high price is at most $(1 - \delta)p_H < p_L$ where the inequality follows from $\delta \geq \delta_h > 1 - (p_L/p_H)$. Thus, the low quality seller would find it more profitable to deviate and charge the low price. A contradiction. We now proceed to prove the existence of a fraudulent hybrid equilibria outcome characterized by $\phi_H^* = 1$, $\phi_L^* = \underline{\phi}(\delta, \pi, p_H)$, $b^*(p_H, h) = 1$ and $b^*(p_H, l) = 1 - (1/\delta)\,(1 - (p_L/p_H))$ for all $\delta \in [1 - (p_L/p_H), \delta_h)$. By Bayes' consistency along the equilibrium path, the buyer's posterior beliefs $\mu^*(H|p_H, s; \underline{\phi}(\delta, \pi, p_H))$ must be given by Equation (2) for all $s \in S$.

Indeed, $\mu^*(H|p_H, h; \underline{\phi}(\delta, \pi, p_H)) > \mu^*(H|p_H, h; \bar{\phi}(\delta, \pi, p_H)) = (p_H - v_L)/(v_H - v_L)$ for all $\delta < \delta_h$ implying $b^*(p_H, h) = 1$ if $\phi_L^* = \underline{\phi}(\delta, \pi, p_H)$ and $\delta < \delta_h$. The low-quality seller's expected revenue if he charges the high price is given by $((1 - \delta) + \delta b^*(p_H, l))p_H = p_L$. Hence, the optimal response by the high-quality seller is to charge the high price. Characterizations A6–A7 complete the proof of the existence of fraudulent hybrid equilibria for all $\delta \in (1/2, 1)$. This completes the proof of Lemma A5. □

The proof of Proposition 1 is completed. □

**Proof of Proposition 2.** If $p_H \leq 2p_L$ then $1 - (p_L/p_H) \leq 1/2$ and the result is immediate by the proofs of Lemmatta A1–A2, A5 and A7, and the fact that $\underline{\phi}(\delta, \pi, p_H)$ is strictly decreasing in $\delta$. If $2p_L < p_H \leq \bar{v}$ then only fraudulent pooling equilibria can be supported if $\delta < 1 - (p_L/p_H)$ by Lemma A1 and only fraudulent hybrid $L$ can be supported in equilibrium for $\delta > 1 - (p_L/p_H)$ by Lemma A2. If $\delta = 1 - (p_L/p_H)$, then a continuum of equilibrium outcomes characterized by $\phi_H^* = 1$, $\phi_L^* \in [(\pi/(1 - \pi))(p_L/(p_H - p_L))((v_H - p_H)/(p_H - v_L)), 1]$, $b_H^*(p_H) = 1$ and $b_L^*(p_H) = 0$ can be supported in equilibrium with buyer's posterior beliefs $\mu^*(H|p_H, s)$ given by Equation (2) for all $s \in S$. The result is then immediate. Under the remaining conditions, only fraudulent hybrid $H$ can be supported in equilibrium for any $\delta < \min\{\delta_h, 1 - (p_L/p_H)\}$ by the proof of Characterization A6 whereas only fraudulent hybrid $L$ can be supported in equilibrium for any $\delta > \max\{\delta_h, 1 - (p_L/p_H)\}$ by the proof of Characterization A7. The result follows trivially from $\bar{\phi}(\delta, \pi, p_H)(\underline{\phi}(\delta, \pi, p_H))$ being strictly increasing (decreasing) in $\delta$. □

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
