# Peer review of "Signalling, Information and Consumer Fraud"

_games, doi:10.3390/g11030029_

Round 1

Reviewer 1 Report

A large body of literature, particularly in the fields of marketing and economics, investigates whether prices can be used a signal of product quality to consumers. Most of these papers assume consumers are either exogenously informed or completely uninformed about the product quality. Relative to the literature, this paper has a somewhat novel setup — it assumes each consumers possesses a private signal of the actual quality. Such a private signal is informative but imperfect. Thus, it presents a two-sided asymmetric information model.

The setup of the model is quite close to that of Voorneveld et al. (2011), with the key differentiation that this paper focuses on pooling and mixed strategies whereas the latter focuses on pure strategies. As a result, its findings are quite different. For example, whereas Voorneveld et al. show honest reporting prevails in the market, this paper shows fraud necessarily arises and exists fro all parameter values. It also finds that a more precise private signal on the customer's end can lead to more prevalent fraud and lower welfare.

Overall, I think the paper studies a sufficiently interesting problem and its findings have important policy implications. The author(s) will be happy to learn that I have recommended a minor revision.

Specifically, my minor suggestion concerns the following. I am okay with the assumption that each consumer has an endowed private signal without incurring any cost of information acquisition. At the same, given the breadth of the extant literature, it may be useful to briefly discuss (perhaps qualitatively) what the equilibrium may be different if the consumer has an option to acquire a more precise signal at a fixed cost. Related to this point, the literature has looked at the case of a perfect "testing" signal that perfectly reveals the state, see https://doi.org/10.1287/mksc.2019.1201 and https://doi.org/10.1016/S0167-7187(00)00057-6 for examples). Of course, the consumer does not have to acquire this signal, but it would be interesting to discuss how the presence of this "testing" option might change the equilibrium. In the presence of this option, would it still be the case that a more precise signal can lead to more prevalent frauds? Along the same line, it may be useful to discuss what actions the seller can undertake to improve the chance of trade. 

Wording and style issues:

  • In the abstract, "a more precise customers’ private information" should be "a higher precision of the consumer's private information"

  • Page 2: "his successfulness" should be "his probability of success"

  • Page 8: "Due to the informativeness of the private signal, the posterior beliefs upon receiving a high (low) signal realization and a high price offer are increasing (decreasing) in the level of precision. Due to the informativeness of the private signal, the posterior beliefs upon receiving a high (low) signal realization and a high price are increasing (decreasing) in the precision level of the signal."
    These are duplicative sentences and should be corrected.

  • Page 12: Please explain a bit more about the following: "In the fraudulent hybrid H equilibrium, only the buyer who observes the high signal is the targeted customer. Therefore, the more precise the information, the more frequently the high-signal buyer must accept trade in order to keep the low-quality seller indifferent between charging either price." The current explanation may be quite confusing to the uninitiated. 

  • Page 13: Relevant to the rather counterintuitive message ("Consumer Protection Agencies concerned about consumers’ welfare and about protecting them from unfair commercial practices should not favour more accurate private information provision policies as long  as this provision is relatively imprecise"), what actions should consumer protection agencies undertake instead?

Reviewer 2 Report

The paper investigates the effect of private and public information precision on the incidence of fraud and welfare in monopolistic & monopsonistic experience goods markets. The main results of the paper are noteworthy, including non-monotonicity of fraud in consumers' private information (although this has already been shown for credence goods by Kerschbamer et al. (2019) and Rigos et al. (2019))

Below I outline several suggestions on improving the presentation of those results:

  1. It would be great to open up the abstract by mentioning the type of your model. This would clarify to the reader which scenarios your findings apply to. E.g. "In a two-sided asymmetric information market...."
  2. 2. Your examples of sea food mislabelling and fraudulent COVID-19 remedies seem relevant for your model, however it would be even better if you could also link your policy recommendations back to real-life asymmetric information markets, possibly with examples. How exactly will the regulator be able to tell which equilibria are currently observed (since welfare-increasing policies crucially depend on that)? How can they decrease/increase the precision of consumers' private signals? Do your policy recommendations regarding "public signal" (price) only apply to markets with price controls (line 455)? 
  3. Line 328: please explain why you think that is the most interesting parameter configuration.
  4. Lines 328-346: Please rewrite these two paragraphs, structuring your findings for different parameter values around an overarching statement, such as non-monotonicity of fraud w.r.t. signal precision.
    Please provide intuitive interpretations of the parameter conditions, e.g. "the prior falls short of the buyer's threshold posterior for accepting the high price" etc.
  5. line 372: do you mean "generic" rather than "non-generic"? If you truly mean non-generic, are these equilibria worth considering?
  6. lines 376-385: This is an example of a clear, well laid-out verbal statement of a result. I wish more results in this paper were laid out in the same manner.
  7. section 6.2: please provide intuition for these results
  8. line 575: you say that incidents of fraud "become strictly positive" - it would be more clear if you started that sentence with where they start, e.g. they are zero for low levels of delta.
  9. Most of your comparative statics results stated in terms of parameter values would greatly benefit from a clear intuition, presented in prose. I highly recommend it.

Typos / language suggestions:

  1. lines 284-287: repeated sentence, please delete
  2. line 292: surely \phi_L denotes the low seller's mixed strategy? It can ALSO be interpreted as the precision of prices as signals of quality. "Corresponds to" seems to convey this better than "denotes"
  3.  line 355: "gains FROM trade" is a much more accepted term than "gains of trade"
